# Nutrition, Sleep, and Exercise as Healthy Behaviors in Schizotypy: A Scoping Review

**DOI:** 10.3390/bs12110412

**Published:** 2022-10-26

**Authors:** Keri Ka-Yee Wong, Adrian Raine

**Affiliations:** 1Department of Psychology and Human Development, University College London, London WCH1 0AA, UK; 2Departments of Criminology, Psychiatry, and Psychology, University of Pennsylvania, Philadelphia, PA 19104, USA

**Keywords:** schizotypy, nutrition, sleep, exercise, behaviors, lifespan, scoping review

## Abstract

This scoping review identifies the role of nutrition, sleep, and exercise as healthy behaviors in non-clinical individuals with schizotypy throughout the lifespan. Methods: We systematically reviewed the existing literature on these topics through databases including: *PsycINFO*, *Scopus*, *APA PsycNet*, *ScienceDirect*, *Wiley Online Library,* and *SpringerLink*. Results: Of the 59 studies found, a total of 29 studies met the inclusion criteria on the review topic. Included studies reflect varying study designs (cross-sectional, multiple time-point, intervention, randomized-placebo controlled trials), assessment of schizotypy and associated healthy behaviors, focus on various samples and lifespan (e.g., undergraduates, adolescents, at-risk individuals), and stem from different countries. Conclusion: While a moderate number of studies address the role of nutrition, sleep, and physical exercise in relation to schizotypy, studies intersecting these topics are limited. Of the limited studies that do exist, the majority are correlational with the beginnings of causal support from intervention studies. As such, more research is needed on the topics of nutrition, sleep, and exercise in relation to schizotypy. Specifically, future research should focus on providing a more holistic understanding of schizotypal traits and its subtypes, and which specific or combination of behaviors may reduce levels of schizotypy.

## 1. Introduction

Schizophrenia is characterized as a multidimensional disabling disorder affecting approximately 1% of the general population [1]. It is globally ranked as the most costly mental disorder to society, with an approximated cost of USD$13,256 per patient [2]. Compared to the general population, individuals diagnosed with schizophrenia also tend to have high rates of physical illness, cardiovascular and respiratory diseases, and suicide rates [3]. Some believe that an individual’s poor physical health is due to the subscription of antipsychotic medications, while others believe that schizophrenia symptoms (namely suspiciousness and disorganization) can promote sedentary behaviors and reduce outward interactions with others [4]. While a host of biopsychosocial causes may be related to schizophrenia itself [5,6], a better understanding of the literature on “healthy behaviors” and ways to treat and prevent this disorder has particular clinical and research interest, as to date, no single intervention exists.

One area of research that has been shown to be of promise is research into schizotypal personality disorder (SPD) and its associated biopsychosocial causes. SPD is a multidimensional personality disorder thought to reflect liability for schizophrenia and other psychotic disorders [7]. Although worthy of study in and of its own right, research into SPD can also offer an effective theoretical framework for understanding the etiology of schizophrenia [8]. In the *Diagnostic and Statistical Manual Fifth Edition* [9], SPD is largely classified into *positive* (e.g., ideas of reference, odd beliefs, unusual perceptual experiences, suspiciousness), *negative* (no close friends, constricted affect, odd behavior), and *disorganized* (odd thought and speech) features often seen as a milder and more attenuated form of schizophrenia. Common correlates associated with individuals with schizophrenia-spectrum disorder and SPD are poor nutrition [10], poor sleep [11], and reduced physical activity [12]. SPD has been found to exist developmentally on a continuum of severity spanning from severe clinical patient populations (clinical SPD) to milder more attenuated forms in non-clinical community samples (psychometrically-defined schizotypy), hence a review of the literature including all of these states from milder attenuated states to more disabling clinical states can have valuable research potential in advancing our understanding of the etiology of schizophrenia and informing early assessments and preventive interventions [13].

Overall, the study of schizotypy and its biopsychosocial causes are well detailed in several independent reviews on the topic [13,14]. However, a review focused on the role of nutrition, sleep, and physical exercise as healthy behaviors for individuals with schizotypy in non-clinical samples, including the overlapping findings across behaviors is non-existent. This review is important as it can inform our understanding of the causes of, and potential interventions for, schizotypy in the general population well before an individual may develop more severe, unchangeable symptoms. Thus, taking a developmental lifespan approach, the current scoping review—chosen as supposed to a systematic or meta-analytic review because of the number of available studies on the topic—aims to address the research question: What is known from the existing literature on the relationship between nutrition, sleep, and physical exercise as healthy behaviors in non-clinical populations with schizotypy? Drawing from studies of non-clinical samples across ages, we include both published work, theses, work under review, and interventions on this topic to discuss treatment implications before concluding with an overview of the gaps and future directions in this area of research.

## 2. Materials and Methods

The main purpose of this review is to explore and understand the extent to which absence of healthy behaviors (nutrition, sleep, and exercise) affect the onset and development of schizotypal personality traits throughout the lifespan.

The search strategy following Arksey and O’Malley’s [15] 5-step framework for scoping reviews involved: (1) identifying the research question, (2) identifying relevant scholarly articles, (3) study selection and screening, (4) charting the data and, (5) collating, summarizing, and reporting findings. To address our research question, ‘What is known from the existing literature on the relationship between nutrition, sleep, and physical exercise in non-clinical populations with schizotypy?’ (Step 1), we searched and included all available papers published in English, across databases, reference lists of relevant studies, and key authors in the fields to ensure the inclusion of the latest unpublished papers relevant to the research question with the latest search date being 27 September 2022. We also included studies with tangential search terms related to schizophrenia-spectrum disorder, psychosis, schizophrenia, and psychotic-like experiences so as to not miss any potential studies with inclusion of schizotypy in their sample. The databases included: *PubMed, Web of Science,/Medline, PsycINFO*, *Scopus*, *APA PsycNet*, *ScienceDirect*, *Wiley Online Library*, *Google Scholar*, and *SpringerLink* with no limitations placed on publication date. We followed the PRISMA-ScR guidelines and checklist [16], which can be found online (https://osf.io/h8356/, accessed on 31 August 2022).

Initially, the terms “schizotypy” and “schizotypal” were used and it was soon apparent that a large proportion of the publications deviated from the original research focus on the more widely researched areas of “schizophrenia” and “psychosis”. The terms “Schizotypal Personality Disorder” and its initials SPD were, therefore, introduced to refocus the search to its original topic. Subsequently, the following terms were searched in relation to the schizotypy variations: *sleep, sleep quality, rest, dream, nightmare, insomnia, exercise, physical activity, sport, movement, workout, CrossFit, running, sedentary, training, gym, athletics, fitness, healthy, nutrition, food, nutrients, diet, calories, fruits, vegetables, eating habits, eating disorder, obesity, weight, nourishment, malnourishment*. Different combinations of these search terms were used in the various search databases to identify frequently cited relevant articles and sources. Reference lists and bibliographies of the relevant articles were also examined to identify the secondary literature. Lastly, *Elicit* (elicit.org, accessed on Accessed on 1 March 2022)—a recent A.I. tool developed to synthesize the findings of research questions—was also used to identify further articles to be included in the review. Using this tool, four additional results were retrieved of which two relevant papers were included.

This process yielded 59 articles, which were subsequently grouped into three main areas: nutrition, sleep, and physical exercise (Figure 1). Articles that explored more than one of these healthy behaviors were included in both groups (see Figure 2) and data were extracted from each of the papers by two independent raters and discussed to address discrepancies to determine study relevance (Table 1). After a further synthesis between two raters, two articles were excluded due to not being accessible and 28 articles were discarded for not fitting the purpose of the review (n = 11), with studies focusing on psychosis/schizophrenia-spectrum disorders and/or not acknowledging schizotypy or schizotypal traits (n = 17) in the study being excluded. The total number of included studies in the review is 29 articles.

## 3. Results

The set of studies reviewed are separated into relevant subsections and summarized below. As some studies cover more than one topic, these are reviewed in all relevant sections.

### 3.1. Nutrition & SPD

Poor dietary habits and nutritional intake have been identified as risk-factors in need of intervention in those with psychosis and schizophrenia-spectrum disorders, with interventions early in development being shown as particularly effective [17]. A review of the literature on the nutrition–schizotypy relationship returned three experimental studies with a focus on nutritional enhancement, particularly focused on the role of omega-3 polyunsaturated fatty acids (PUFAs) as the active treatment.

In Vienna, Amminger et al. [18] conducted a randomized, double-blind, placebo-controlled trial investigating the role of omega-3 PUFAs supplements on slowing the progression to first-episode psychotic disorders and symptom reduction in adolescent patients aged 13 to 25 years old. A total of 81 patients deemed as ultra-high risk with a diagnosis of schizophrenia-spectrum disorders and schizotypal personality disorder were randomly assigned to either the treatment condition (n = 40), where they received a daily dose of approximately 1.2-g omega-3 PUFAs or the placebo group (n = 40) who received a daily coconut oil capsule containing vitamin E and 1% fish oil to match the appearance and flavor of the treatment capsule. In total, 76 participants completed the trial (93.8%). Compared with the placebo group, treatment group patients were significantly less likely to convert to psychosis at 12-month follow-up, 4.9% vs. 27.5%. Throughout the treatment, patients who received omega-3 PUFAs reported significantly better global functioning compared to the control group and reductions across symptoms with moderate (negative symptoms) and moderate to large effects (positive, general, total symptoms). One patient treated with omega-3 did develop psychosis during the posttreatment follow-up. The reduction in prodromal symptoms and functioning in the treatment group was sustained even after the intervention stopped. These findings suggest a positive potential behavioral response to omega-3 supplementation in intervening and slowing down symptom progression in young people at ultra-high risk for developing schizophrenia.

These results failed to replicate in a large double-blind placebo-controlled randomized clinical trial by McGorry et al. [19]. In this study, 304 patients with ultra-high risk for psychosis either received 1.4 g of omega-3 PUFAs or a placebo (paraffin oil), in addition to 20 or fewer sessions of high-quality psychosocial intervention (cognitive behavioral case management and antidepressants) over a six-month study period. Conversion to psychosis status at six months for the intervention and control group was 6.7% and 5.1%, and at 12 months, 11.5% and 11.2%, respectively, suggesting no significant treatment effect of omega-3 PUFAs in reducing transition to psychosis. However, symptom and functional improvements were observed more broadly for both groups. The authors concluded that omega-3 supplements may not have any added value in patients who are already receiving high-quality psychosocial treatment and antidepressants to help them cope with their condition.

Conversely, another study by Raine et al., [20] testing the effects of environmental enrichment program consisting of nutritional supplement (where treatment children received 2.5 portions of fish more than the control group), exercise, and education was trialed in a large cohort of children from the general population in Mauritius. In total, 100 children were selected through stratified random sampling to be in the intervention group from a representative sample of 1795 children born in Mauritius between 1969 and 1970 and were aged 3 at the time of recruitment and were compared to 355 children selected from the remaining 1695 to be in the control group. Between 3 and 5 years old, children in the environmental enrichment treatment group were in preschools with a teacher–pupil ratio of 1:5.5 where children received *nutrition* (milk, fruit juice, hot meal, fish, chicken or mutton, and salad at school), *physical exercise* (average of 2.5 h of physical activities, field trips, walking, gymnastic classes, structured outdoor play, free play), a *nap* after lunch, and *education* (skills training in verbal, visuospatial coordination, conceptual skills, perception, memory, and sensation) compared to children in the control group who had traditional Mauritian grade-school curriculum experience at *petites ecoles* (Dame schools) with no milk, structured exercise, with lunch that was typically bread-only, rice and bread, or rice only, and a teacher–pupil ratio of 1:30. Of the 100 children in the enrichment program, full behavioral outcome data were available at age 17 (n = 83) and 23 years (n = 73), which allowed for comparison with the control group (n = 355). This two-year intervention found that there was a significant effect between those in the intervention group compared to the control condition in terms of schizotypy at age 17 years, but not at age 23 years. These findings were especially stark for children who were deemed as malnourished at age 3 years, suggesting that an enriched, stimulating environment is beneficial for psychological and behavioral outcomes even 14 and 20 years later in life.

In discussing their findings, Raine et al. [19] hypothesized that omega-3 supplementation may have contributed to the reductions in schizotypy observed at both ages 17 and 23 in those children who had poor nutrition at age 3 (p. 1632). In 2006, Raine interviewed three of the original staff in Mauritius who had been involved in administering the enrichment, and who were blinded to the 2003 hypothesis. These staff recreated daily food diaries in a typical week of the meals the children in both the intervention and control groups. One key difference observed was that the enriched group received 2.5 child portions of fish more per week compared to the control group who received very little fish at the *petites ecoles* (p. 827) [21]. This suggests, but does not prove, that the increased fish consumption could have been one of the key active ingredients in the enrichment that reduced schizotypy in young adulthood.

Still one other study looked at malnutrition and schizotypy prospectively and along with maltreatment. In a prospective longitudinal study in Barbados, Hock et al. [22] found that malnutrition was associated with increased levels of schizoid and schizotypal personal disorder; however, this relationship was not significant after accounting for maltreatment. This suggests that other important early childhood adversities should be taken into consideration when examining the role of malnutrition and schizotypy.

Taken together, there is some evidence for early nutritional and educational enrichment in reducing psychotic symptoms for at-risk populations and initial evidence for intervention early in development on schizotypy and overall behavioral problems in adulthood, on balance aligns with the early intervention model. Under conditions where patients are already receiving high-quality psychosocial therapy and antidepressants, the added value of omega-3 supplement may not be as large. However, future research drawing on the efficacy of environmental enrichment programs, perhaps extending known interventions for schizophrenia-spectrum disorders and syndromes to attenuated forms of schizotypy, with a combination of diet, sleep, physical exercise, and cognitive therapy/training [17,23] may be particularly useful in informing the field in developing interventions during early development.

### 3.2. Exercise & SPD

Poor physical exercise is another risk-factor shown to be associated with a poor lifestyle and is predictive of poor health, sedentary behaviors, cardiovascular disease, obesity, and other conditions like diabetes, especially for individuals with psychosis and schizophrenia-spectrum disorders (SSD) [24,25,26]. In a large UK Biobank study, individuals with schizophrenia and SSDs (n = 1078) report significantly lower accelerometer ratings (e.g., physical activity as represented by the average vector magnitude, a combination of time and intensity of movement in a week) than controls (n = 450,549), demonstrating high levels of sedentary behavior, which is often not reflected in self-reported measures of physical activity [12]. To date, virtually no studies have explicitly focused on the relationship between physical exercise and schizotypy. With the known mental and physical benefits of exercise, studies in this area are still in their infancy with more room to grow. Of the studies that exist, these tend to focus on increasing the motivation to be more physically active in individual with SSDs and assessing the extent to which physical activity intervention programs may have sustained positive effects on an individual’s lifestyle changes.

Beebe and colleagues have primarily led the way in extending the literature on motivation to exercise in schizophrenia patient populations compared to those with SSDs. Particularly noteworthy are a series of feasibility and RCT studies looking at improving the motivation to exercise in individuals with SSDs. The “Walk, Address, Learn and Cut (WALC)” intervention involves randomly assigning outpatients with SSDs to receive weekly hour-long motivational intervention group session (N = 48) where participants discuss and can ask questions about walking or a time and attention control group session (n = 49) for a month before engaging in the same 16-week walking program [27,28]. This walking program involved warm-up stretches followed by walking that gradually increased in duration (from 5 to 30 min) over the four weeks, and cool-down exercises at the end of each session. Compared with the control group, individuals in the experimental group who received the motivational intervention session prior to the walking intervention attended more walking groups, stayed with the group for longer, and reported on average more minutes walking each month, demonstrating sustained engagement with the activity. However, no measures of symptoms of SSD were assessed and so it is unknown whether increased exercise reduces symptoms.

At 18-month follow-up, comparing patients from the experimental group (n = 11) to the control group (n = 11), the experimental group participants on average still walked significantly more steps and covered more distance on six of the seven days surveyed, demonstrating that participants have integrated walking habits into their lifestyle. These findings were replicated in a smaller feasibility study of 16 patients with SSD engaging in a 10-week walking intervention program based on the social determinant theory of motivation [29]. Authors found that this program increased participant’s weekly walking minutes, quality of life, reduced sedentary behaviors sustained at post-treatment and 1-month follow-up [30]. However, no control group was available for comparison and no assessment of SSD symptom reduction was reported.

The evidence in this section focuses primarily on patients with a diagnosis of SSDs and individuals with a potential diagnosis of schizotypy is a good start but not encouraging as there are no studies on psychometrically defined schizotypy. Whether increased physical activity is beneficial in reducing SSD symptoms is not often reported, and, thus, it is difficult to gauge whether exercise can also reduce symptoms. It does, however, show that by tackling the psychological motivations of an individual through education and a trusting environment for individuals to express their concerns and worries around physical exercise, even for low intensity exercises like walking, coupled with access to a community of like-minded individuals, can lead to sustained behavioral changes that can translate to longer-term lifestyle changes. Given the low economic cost, social gains, and minimal side-effects of exercising, these findings would lend support to social prescribing model of supporting patients with SSDs.

### 3.3. Sleep & SPD

Poor sleep quality is associated with psychotic-like experiences, schizotypy, SSDs, and other mental and physical disorders across the lifespan [31,32,33,34]. Studies in the 1990s found associations between distress caused by nightmares and more schizotypal traits (as well as more positive dreams) in the general population (age range 14–55 years) [35] and frequent nightmares and distress reported by young adults being associated with more schizotypal traits, specifically magical ideation [36], and even predictive of general psychological disturbances (e.g., anxiety, depression, and dissociation) [37,38,39].

An updated review of the literature conducted here on the sleep–schizotypy relationship reveals 10 additional studies of cross-sectional and prospective designs broadly focusing on community-based adolescents and undergraduate students and with limited experimental studies or interventions (in addition to studies included in Koffel and Watson’s 2009 review) [33]. There is also some variation in the assessment of sleep, including the causes of poor sleep as indexed by nightmares, sleep quality, types of sleep, dream quality, and sleep duration to neurobiological signals of sleep, such as sleep spindle densities.

Many of studies have been conducted in undergraduate students and psychometrically defined schizotypy. For example, one two observation study looked at the month in which people were born in relation to levels of schizotypy and the hours of sleep reported [40] and found that those born in the winter/spring (21 December to 20 June) were more likely than those both in the summer/fall (21 June to 20 December) to report higher levels of schizotypy and individuals sleeping less than 6 h or more than 8 h were more likely to score in the top 25% on a schizotypy measure compared to those who slept 6–8 h. In another study of 18–25 years-old, the link between poor sleep and higher general schizotypy scores ranged from 0.31 to 0.45 based on convenient samples, large-scale cross-sectional self-report survey data [41] with specifically strong associations with cognitive disorganization, introvertive anhedonia, and unusual perceptual experiences [42]. Similar findings were replicated in another undergraduate sample (aged 17–24 years) showing unusual sleep disturbances and more schizotypal traits, particularly with magical thinking [43] and still another study looking at dream quality differentially associated with schizotypy subtypes [44]. Researchers have also found that poor sleep hygiene, including emotional rumination before sleep, is associated with higher levels of schizotypy [45], yet other studies have manipulated sleep deprivation (24-h sleep deprivation) and found poorer cognitive performance for high and low schizotypals, but there were no schizotypy group by sleep interaction [46]. In another study, [47] Lustenberger et al. found a relationship sleep spindles density and scores on the Schizotypal Personality Questionnaire (SPQ) in healthy university men in Zurich (N = 20) (*r* = −0.64, *p* < 0.01), but not thalamic glutamate Glx levels. In another study of adolescents (N = 176, Mean = 12.3 years), Kuula et al. [48] found sleep spindle morphology to be associated with a higher proportion of rapid eye movement (REM) sleep—defined as when an individual is having vivid dreams—and high levels of schizotypy. This suggests that lower sleep spindle density is associated with schizotypy in the same way that this association has been found in past studies of schizophrenia patients [49,50,51].

In a more recent study of schizotypy in the adult population in relation to the impact of the COVID-19 pandemic on mental health and sleep [52], Wong et al. [53] surveyed adults aged 18–89 years from the UK, US, Italy, and Greece across three time-points (April–July 2020, October–January 2021, April–July 2021). Using network analysis, researchers found that symptoms of schizotypy were more elevated at time 2 (October–January 2021) compared to times 1 and 3, and at every timepoint they were positively associated with poorer sleep, primarily through changes in the individual’s stress concerning the pandemic and depressive symptoms. However, when the environment changed and the easing of lockdown restrictions took place, participants reported improved sleep quality and frequency, and decreased schizotypal traits across all ages and countries. However, this study did not consider other factors such as dietary habits.

### 3.4. Sleep, Exercise, & SPD

Only one study has examined sleep, exercise, and schizotypy. In a two-timepoint study investigating the impact of covid on schizotypy, exercise, and sleep quality in the UK (n_1_ = 239, n_2_ = 126) and Germany (n_1_ = 543, n_2_ = 401) during May 2020 and October 2020, Daimer et al. [54] found that higher schizotypy scores, especially interpersonal deficits, were being reported in both countries in October 2020 compared to May 2020. Interestingly, lower schizotypy scores were associated with moderate physical exercise, whereby those who exercised five or more times a week reported concurrently lower levels of schizotypy for May 2020 and October 2020. Similarly, they found that individuals who slept 6–8 h or 8 h or more reported lower levels of schizotypy compared with those with under 6 h of sleep, suggesting that more sleep seems to be associated with reduced levels of schizotypal traits.

### 3.5. Nutrition, Exercise, Sleep & SPD

Only one study to date has examined all four aspects of nutrition, physical exercise, and sleep as healthy behaviors in schizotypy, while one other study has examined exercise, nutrition, and schizotypy. In a survey study of US university students (N = 530), Dinzeo and Thayasivam [55] found that increased schizotypy symptoms were associated with poorer sleep quality across positive, negative, and disorganized domains. Specifically, the authors found that individuals scoring high on all three schizotypy domains reported significantly more severe somatic symptoms, poorer psychological health (e.g., reduced engagement with health behaviors), and more sleep difficulties compared to individuals self-reporting as being low across all schizotypy domains. However, the schizotypy group did not differ on the nutrition indicator, which measures engagement with healthy food intake levels. Regarding physical exercise, individuals reporting high and intermediate levels of interpersonal deficits (negative) schizotypy reported significant low levels of exercise, while individuals reporting high on disorganized features were only trending, and no differences were found in exercise levels for those with cognitive-perceptual deficits (positive) of schizotypy.

In another unpublished master’s thesis of university students (N = 115), only the unusual perceptual deficits subscale within the positive symptoms of schizotypy was predictive of healthier behaviors on health and exercise, and overall higher negative symptoms of schizotypy were associated with eating less nutritious foods (e.g., salad) and likelihood of eating out, self-perceived poorer fitness levels relative to peers, and with being less likely to engage in exercising [56]. No relationship was found between negative schizotypy and eating habits.

Taken together, although these findings have design limitations and relationships may be inflated due to single-informant reporting and self-reporting, it is conceivable that low physical exercise engagement may occur prior to the emergence of psychotic disorders, as well as early withdrawals from social situations, which are also reflected in the negative aspects of schizotypy, as demonstrated in studies of non-clinical samples reviewed in this section. Teasing out the causal direction of these constructs would be an important next step in helping inform the timing of interventions and level of support for individuals to achieve better lifestyle changes that may in turn lead to symptom reductions in schizotypy.

## 4. Discussion

The aim of the current scoping review examined the existing literature on the relationship between schizotypy and nutrition, sleep, and physical exercise in non-clinical samples in the general population developmentally. Our results indicate that more rigorous cross-sectional studies with comparable control groups are needed to gauge the significance of group differences in relation to healthy behaviors (nutrition, sleep, and physical exercise). Overall, across the sections, evidence from longitudinal studies is lacking and, thus, the question of “sustained effects” (stability) and the temporal ordering of variables (causal relationships) and their impact on schizotypy have been limited. Virtually no studies are qualitative and so understanding the fundamental reasons *why* these healthy behaviors are related to schizotypy are underexplored. That said, there is initial evidence on nutrition and schizotypy from randomized placebo-controlled trials manipulating nutritional intake and omega-3 supplement levels, but further manipulations of diet/nutrition and exercise levels/intensity would be helpful in teasing apart the comorbid relationships and to move beyond correlational relationship to infer causation.

Evidently, across the sections on clinical patients with SSDs and those on the dimension of schizotypal traits in the general population, there has been huge heterogeneity in the age group of focus impacting the inferences that can be made within the already limited literature. Undergraduate students as convenience sample represents the only studies able to examine all three health behaviors (nutrition, exercise, sleep) and schizotypy together. Whilst this is a good start, this sample limitation clearly impacts the generalizability of findings and future research could perhaps extend to younger ages—adolescence—an important developmental period where a young person begins to negotiate their identity, body image, and self-confidence and when ideas about the self are less fixed and still forming [57]. Research in this developmental period may encourage more healthy ways of thinking and behaving and allow researchers to map the developmental trajectories of how these constructs develop in early development and to inform preventative interventions [13]. In addition, research in early development can also consider interventions with less side-effects as alternatives to medication (e.g., social prescribing) and long courses of therapy that require a certain level of motivation and dedication from individuals [58].

A further research direction concerns evaluating which health behavior is associated with which form of schizotypy. It is generally recognized that at least three dimensions of schizotypy exist—cognitive-perceptual, interpersonal, and disorganized [59]—each with somewhat different underpinnings. Whether different health behaviors are differentially associated with factors of schizotypy is unclear, and future research should report on specific subtypes of schizotypy and how these relate to each of the healthy behaviors. Nevertheless, it is of interest that the latter two sections outlined above found associations specifically with negative features of schizotypy. It has been theorized that interpersonal and disorganized features of schizotypy (“neuro-schizotypy”) are grounded more in neurobiological factors compared to cognitive-perceptual features, which are more based on psychosocial influences (“pseudo-schizotypy”) [14]. It is conceivable, therefore, that healthy behaviors that impact biological functioning may influence the more biologically based features of schizotypy, and future studies are needed to support or refute this speculative hypothesis.

A clear limitation of this scoping review is that it examines a small but growing literature on the intersection of healthy behaviors—nutrition, sleep, physical exercise—and schizotypy and identifies key gaps for future research. For example, there is evidence that the research between sleep and schizotypy is more established than nutrition/exercise and schizotypy—when the latter two behaviors may be more changeable than sleep habits. One recommendation would be that with more rigorous research in these areas of study, specifically nutrition and exercise in relation to schizotypy, as a field, we can begin to inform early lifestyle changes/development. It is also clear in this review that there is a need for a more diverse sample, one that extends beyond undergraduate convenience samples and across cultures, to improve the generalizability of findings. One recommendation would be to co-produce a mixed-method design study of younger participants to inform preventive interventions—perhaps ones that promote lifestyle changes earlier on in development—to better support individuals with schizotypy sooner and more creatively (e.g., through social prescribing or co-produced solution). A better understanding of “what works” and the causal directions of these relationships may also inform support for individuals in later life through improving their quality of life and functioning. A final recommendation for future research would be to pursue more experimental designs where exercise, nutrition, and sleep are manipulated in clinical populations to see whether these healthy behaviors combined with antipsychotic medication can have sustained effects.

In conclusion, our scoping review is a start in highlighting areas of research promise between schizotypy and healthy behaviors (sleep, nutrition and exercise), yet much more is needed to be done in the intersection of these constructs. Taking a developmental cross-cultural lifespan approach, future research should build on existing cross-sectional studies and be of mixed method design co-produced with patients to truly understand the barriers to sustaining healthy behaviors, as well as incorporate assessments of multiple healthy behaviors in relation to schizotypy, to truly understand what works in preventing schizotypal traits in later life and in different populations.

## Figures and Tables

**Figure 1 behavsci-12-00412-f001:**
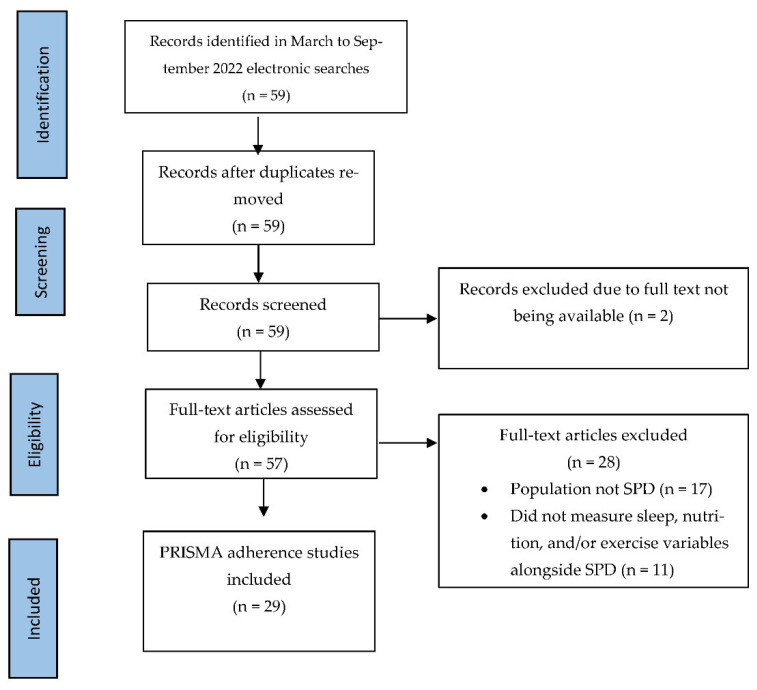
Flowchart of literature search of the number of studies on schizotypy and sleep, nutrition, and physical exercise following the PRISMA-SCR guidelines.

**Figure 2 behavsci-12-00412-f002:**
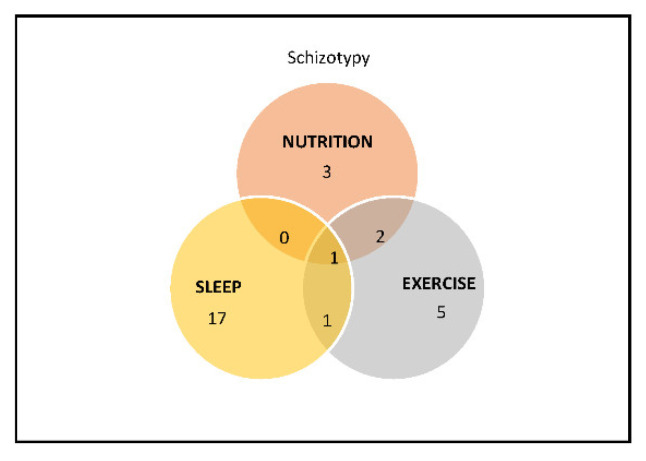
Venn diagram of studies of schizotypy and its overlap in different topics.

**Table 1 behavsci-12-00412-t001:** Details of all studies meeting exposure (nutrition, sleep, exercise) and schizotypy (SPD, schizotypy, schizotypal personality disorder).

Author(s)	Year	Exposure Variable	Origin (Where the Study Was Published or Conducted)	Aims/Purpose	Study Population	Sample Size	Methods	Outcomes Measures	Key Findings in Relation to Review Question
Amminger, Schäfer, Papageorgiou., et al.	2010	Nutrition	Vienna, Austria	Efficacy of omega-3 supplement in reducing/slowing the conversion to psychosis in ultra-high-risk patients.	Adolescent ultra-high-risk patients diagnosed with schizophrenia-spectrum disorder (SSD) and schizotypy aged 13 to 25 years old.	N = 81 Treatment n = 40 vs. placebo n = 40 (F = 53, M = 27)	Randomized, double-blind, placebo-controlled trial investigating the role of omega-3 PUFAs supplements on slowing the progression to first-episode psychotic disorders and symptom reduction in patients. A total of 81 patients deemed as ultra-high risk with a diagnosis of schizophrenia-spectrum disorders and schizotypal personality disorder were randomly assigned to either the treatment condition (n = 40), where they received a daily dose of approximately 1.2-g omega-3 PUFAs or the placebo group (n = 40) who received a daily coconut oil capsule containing vitamin E and 1% fish oil to match the appearance and flavor of the treatment capsule.	Conversion to psychosis (PANSS), global functioning	In total, 76 participants completed the trial (93.8%). Compared with the placebo group, treatment group patients were significantly less likely to convert to psychosis at 12-month follow-up, 4.9% vs. 27.5%. Patients who received omega-3 PUFAs reported significantly better global functioning compared to the control group and reductions across symptoms with moderate (negative symptoms) and moderate to large effects (positive, general, total symptoms). The reduction in prodromal symptoms and functioning in the treatment group was sustained even after the intervention stopped. These findings suggest a positive potential behavioral response to omega-3 supplementation in intervening and slowing down symptom progression in young people at ultra-high risk for developing schizophrenia.
Barnes, Koch, Wilford and Boubert	2011	Sleep	UK	An investigation into personality, stress, and sleep with reports of hallucinations in a normal population	Oxford Brookes university students.	N = 117 (F = 75, M= 42)	Self-report survey data; cross-sectional	sleep (Pittsburgh), schizotypy (PRE-MAG)	Cognitive Disorganization (r = 0.365), Introvertive anhedonia (r = 0.305), and Unusual Experiences (r = 0.239).
Báthori, Polner and Simor	2022	Sleep	Hungary	A three-week long study to examine the bidirectional, temporal links between dream emotions and daytime PLEs, taking into consideration schizotypal personality traits	Non-clinical individuals with high dream recall	N = 55 (F = 42, M = 13)	Self-report survey data, cross-sectional	Schizotypy (O-LIFE) Sleep (the “Morning Questionnaire”, an 8-point single item survey where participants rated the quality of their sleep)	General Disorganized schizotypy predicted negatively valanced dream emotions. Dream emotions were both predicted by dispositional schizotypal traits and by psychotic-like experiences as measured the night before sleep.
Beebe & Smith	2010	Exercise	South-eastern USA	Assessing the feasibility of the Walk, Address, Learn, and Cue (WALC) intervention for schizophrenia spectrum disorders	People with SSDs receiving care at an outpatient treatment facility	N = 17 (F = 7, M = 10; *M_age_* = 43.2 years, range = 24–54 years; )	Randomly assigned outpatients with SSDs who are receiving care at a treatment facility to receive weekly hour-long motivational intervention group session (N = 17). Most of the participants were male Caucasians with schizoaffective disorder and on antipsychotic medication.	Demographic data were collected at study entry and included: age, race, gender, living arrangement, chart diagnosis, and self-reported educational level. Attendance at WALC-S groups was defined as the percentage of sessions attended out of the total sessions offered. Reasons for nonattendance were obtained during follow-up telephone calls after each WALC-S group.	Compared with the control group, individuals in the experimental group who received the motivational intervention session prior to the walking intervention attended more walking groups, stayed with the group for longer, and reported on average more minutes walking each month, demonstrating sustained engagement with the activity. However, no measures of symptoms of SSD were assessed and so it is unknown whether increased exercise reduces symptoms.
Beebe, Smith, Burk, et al.	2011	Exercise	Southeastern USA	Aimed to look at the effect of a motivational intervention on exercise behavior in persons with schizophrenia spectrum disorders.	outpatients with SSDs	N = 97 (treatment = 48 vs. controls = 49; average age = 46.9 years (SD = 2.0) (F = 46, M = 51)	Data collection at 2 years post waking intervention program.	Exercise (Walking group attendance, persistence, compliance)	Treatment group reported higher levels of exercise attendance, persistence and compliance compared to controls. WALC-S recipients attended more walking groups, for more weeks and walked more minutes than those receiving TAC. Percent of WALC-S or TAC groups attended was significantly correlated with overall attendance (r = 0.38, *p* = 0.001) and persistence (r = −0.29, *p* = 0.01), as well as number of minutes walked. This study is among the first to examine interventions designed to enhance exercise motivation in SSDs.
Beebe et al.	2013	Exercise	Southeastern USA	To examine the sustained impact of an exercise intervention on individuals with SSD at 14 and 34 months post-RCT program	Followed-up individuals with SSDs who had taken part in an RCT exercise intervention to see how they fair at 14–34 months	N = 22 (23–71 years-old; 11 = exp, 11 = control) (F = 11, M = 11)	Measured participants’ step count and distance covered using a pedometer	Exercise (steps and distance using pedometers)	Found that the experimental group participants walked more steps and cover more distance on average than control participants on 6 of the 7 days. Physical activity level of persons with SSDs after exercise intervention. <5000 steps daily = sedentary; 7000–13,000 steps/daily = healthy.
Claridge, Clark, and Davis	1998	Sleep	UK	Nightmares, dreams, schizotypy	Community-residing 14–55-year-olds	(N = 204) convenience sample (M = 87, F = 117)	Self-report survey data, cross-sectional	Schizotypy (STA), age, sleep (Belicki’s Nightmare Distress scale)	Found that higher levels of nightmare distress are associated with more schizotypy, but interestingly also more vivid and enjoyable dreaming (attributed to greater imagination abilities in magical thinking).
Daimer, Mihatsch, Ronan, et al.	2021	Sleep and Exercise	Germany & UK	Investigated the impact of the COVID-19 pandemic on mental health with a specific focus on schizotypal traits in the general population of Germany and the UK in April/May vs. September/October 2020	Independent samples in the UK and Germany during May 2020 and October 2020	UK (N = 239, N = 126) Germany (N_1_ = 543, N_2_ = 401) Gender distribution: first survey (F = 72%, M = 25%, undefined: 3%), second survey (F: 69%, M: 25%, undefined: 6%)	Two independent samples from the UK and Germany self-reporting on a survey at two time-points	Schizotypy (SPQ), exercise (days with physical exercise per week) and sleep (hours per night) quality measured by hours	Found that higher schizotypy scores, especially F2, in October compared with May 2020. Moderate physical exercise was associated with decreases in SPQ scores. Those who exercised 5 or more times a week reported concurrently lower levels of schizotypy for May and October 2020. Those who slept 6–8 h and 8 h+ reported lower levels of schizotypy compared with those with under 6 h of sleep.
Dinzeo and Thayasivam	2021	Sleep, Nutrition, and Exercise	USA	Investigated the relationship between schizotypy, lifestyle behaviors, and health in a young adult sample	University students	N = 530 (F = 48.6%, M = 51.4%)	Self-report survey data, cross-sectional	Schizotypy (SPQ-B), exercise (LHQ-B), sleep quality (PSQI)	Increased schizotypy symptoms were associated with poorer sleep quality across positive, negative, and disorganized domains. Individuals scoring high on all three schizotypy domains reported significant more severe somatic symptoms, poorer psychological health (e.g., reduced engagement with health behaviors), and more sleep difficulties compared to individuals self-reporting as being low across schizotypy domains. The schizotypy group did not differ on the nutrition indicator, which measures engagement with healthy food intake levels. Regarding physical exercise, individuals reporting high and intermediate levels of interpersonal deficits (negative) schizotypy reported significant low levels of exercise, while individuals reporting high on disorganized features were only trending, and no differences were found in exercise levels for those with cognitive-perceptual deficits (positive) of schizotypy.
Faiola et al.	2018	Sleep	Germany	Replicating previous findings of cognitive deficits in high (as compared to low) schizotypy and after sleep deprivation (as compared to normal sleep), and the group by sleep interactions	Healthy subjects with high or low levels of positive schizotypy	N = 36 (out of 5006 completed survey) (high schizotypy (≥1.25 SD) = 17 vs. low schizotypy(≤0.5 SD) = 19) (F = 24, M = 12)	Sleep report survey and sleep deprivation intervention	Schizotypy (O-LIFE) Sleep (sleep deprivation intervention)	Sleep deprivation impaired performance in the go/no-go and n-back tasks relative to the normal sleep control condition, suggesting that sleep deprivation had implications on inhibitory control and potentially, overall attention to task rather than working memory deficits. However, no differences were found between groups or interaction with sleep conditions, suggesting no cognitive impairments across schizotypy groups. This is perhaps due to the small sample size and strength of the sleep deprivation condition (more than 24 h deprivation needed).
Firth et al.	2018	Exercise	UK	Using large-scale population-based UK Biobank study data, to examine the difference in objective and subject measures of physical activity.	Individuals with schizophrenia and SSDs	N = 1078 (F = 45%, M = 55%)	Self-report survey data, accelerometer data, cross-sectional	Physical activity (accelerometer) (IPAQ), Schizophrenia-spectrum disorder status	SSD individuals had significantly lower accelerometer ratings (e.g., physical activity as represented by the average vector magnitude, a combination of time and intensity of movement in a week) than controls (n = 450,549), demonstrating high levels of sedentary behavior, which is often not reflected in self-reported measures of physical activity.
Hock et al.	2018	Nutrition	Barbados	To examine the unique and combined associations of exposures to early malnutrition and childhood maltreatment with PD scores in middle adulthood	Children born between 1967 and 1972 who had experienced moderate to severe protein-energy malnutrition in their first year of life	N = 139 (malnutrition = 77 vs. control = 62; M_age_ = 43.8, SD = 2.3 years) (F = 66, M = 73)	Self-reported survey, structured interviews. Participants were split into a previously malnourished group and a control group. They compared mean scores of personality pathology by malnutrition history group. They also ran linear regression analyses with malnutrition histories as independent predictor variables and adult personality pathology as dependent variables.	Assessments at 40–45 years: Early malnutrition (protein-energy marasmus in the first year of life) Personality disorder (NEO FFM PD, SCID-II-PQ)	Malnutrition positively correlated with SPD but not significant after adjusting for childhood maltreatment and standard of living.
Knox & Lynn	2014	Sleep	New York, USA	To evaluate the contextual independence and the generalizability of the relation between sleep-related experiences and a variety of measures (including schizotypy) found to correlate with such experiences when assessed in the same context	Undergraduate students at the State University of New York.	N = 173 86 participants in the out-of-context condition vs. 87 in the in-context condition (F = 106, M = 67; Median age = 18 years; range = 17 to 24 years)	Self-reported surveys. Participants completed the measures over two separate 1-h experimental sessions—a “personality” session and a “sleep” session.	Sleep (ISES, PSQI) Schizotypy (PER-MAG, REF, SAS)	Sleep experiences were associated with measures of dissociation, absorption, and schizotypy. Of the schizotypy measures, magical ideation is the strongest correlate of unusual sleep experiences. Observed relations between sleep experiences and measures of schizotypy are unaffected by variations in the experimental context and appear to be generalizable in this regard.
Koffel and Watson	2009	Sleep	USA	Extensive review on the relationship between unusual sleep experiences, dissociation, and schizotypy in both clinical and non-clinical samples	studies	N/A	Review paper	Sleep (nightmares, vivid dreaming, narcolepsy symptoms, and complex night time behaviors), dissociation, schizotypy (PANAS)	Found that unusual sleep experiences (e.g., nightmares, vivid dreams, narcolepsy symptoms, complex behaviors at night) are associated with symptoms of dissociation/schizotypy in both clinical and non-clinical samples. Evidence that unusual sleep experiences, dissociation, and schizotypy are more strongly related to each other than to other measures of daytime symptoms (e.g., negative affectivity/neuroticism, depression, anxiety, substance use) and sleep complaints (e.g., insomnia and lassitude).
Kuula, Merikanto, Makkonen, et al.	2019	Sleep	Helsinki, Finland	Investigated schizotypal traits, sleep spindles, and rapid eye movement in adolescence	Adolescents (urban community-based cohort composed of 1049 healthy singletons born between March and November 1998 in Helsinki, Finland)	N = 176 (Mean = 12.3 years) (F = 61%, M = 39%)	Behavioral and self-reported survey; cross-sectional	Sleep spindle morphology (ambulatory overnight polysomnography), REM sleep (EEG), schizotypy (SPA)	Found sleep spindle morphology to be associated with a higher proportion of rapid eye movement (REM) sleep—defined as when an individual is having vivid dreams—and high levels of schizotypy.
Levin	1998	Sleep	USA	Iinvestigated the relationship between nightmares and schizotypy	university female students; frequent nightmare group with at least 1 weekly nightmare	N = 60 (30 recurring nightmare group, 30 control group)	Self-report survey data; cross-sectional	Schizotypy (Chapman scale, SSP), depression (BDI), anxiety (STAI), nightmare (1 page description)	Found that those with DSM III SPD diagnosis reporting 1+ nightmares/week reported significantly more schizotypal traits, specifically magical ideation, compared to those who report no nightmares in the same duration.
Levin & Raulin	1991	Sleep	USA	Investigated the relationship between frequent nightmares and schizotypal symptomatology	University students	N = 669 (F = 446, M = 223)	Self-report survey data; cross-sectional	Schizotypy (four scales measuring perceptual aberration, intense ambivalence, and somatic symptoms. The origin of these scales can be found on p. 9) Sleep (nightmare frequency checklist), sex	Found a positive relationship between nightmare frequency and 3 (perceptual aberration, intense ambivalence, and somatic symptoms) of 4 schizotypy measures. Observed relationships stronger for females than for males. There was a negative relationship between physical anhedonia and nightmare frequency.
Levin & Fireman	2002	Sleep	USA	Nightmare prevalence, nightmare distress, and self-reported psychological disturbance	University students	N = 116 (mean age = 20 years) (F = 85, M = 31)	Self-report survey data; cross-sectional	Sleep (frequency and distress—21-day dream logs with Likert-rating scales), schizotypy (SPQ)	Found that those who reported 3+ nightmares/week reported significant more schizotypy traits than those who reported 2 nightmares/week. It is nightmare distress not frequency that is associated with psychological disturbances like anxiety/depression/dissociation (r = 0.31–0.55). Nightmare distress was also associated with prevalence of nightmare reported in diary logs. Limitation of study does not include childhood trauma.
Lustenberger, O’Gorman, Pugin, et al.	2015	Sleep	Zurich, Switzerland	Sleep spindles are related to schizotypal personality traits and thalamic glutamine/glutamate in healthy subjects	Zurich University healthy men	N = 20 (age = 23.3 ± 2.1 years)	Self-report survey and 2 all-night sleep electroencephalography recordings (128 electrodes).	Sleep spindle densities (EEG), schizotypy (SPQ)	Found a relationship between sleep spindles density and scores on the Schizotypal Personality Questionnaire (SPQ) in healthy university men in Zurich (N = 20) (*r* = −0.64, *p* < *0*.01), but not thalamic glutamate Glx levels.
Ma, Zhang, & Zou	2020	Sleep	Guangzhou, China	The mediating effect of alexithymia on the relationship between schizotypal traits and sleep problems among college students	First year university medical students	N = 2626 (18–25 years old) (F = 1601, M = 1025)	Self-report survey data; convenience sample; cross-sectional	Sleep (ISI), schizotypy (SPQ)	Found a link between poor sleep and higher general schizotypy scores range from 0.31 to 0.45 based on convenient samples, large-scale cross-sectional self-report survey data.
McGorry, Nelson, Markulev, et al.	2017	Nutrition	Australia, Asia, Europe	Efficacy of omega-3 supplement in reducing/slowing the conversion to psychosis in ultra-high-risk adolescents	Patients with ultra-high risk for psychosis either received 1.4 g of omega-3 PUFAs or a placebo (paraffin oil), in addition to 20 or fewer sessions of high-quality psychosocial intervention (cognitive behavioral case management and antidepressants) over a 6-month study period.	N = 304 (age = 13–40 years) (F = 165, M = 139) 153 ω-3 PUFAs vs. 151 placebos	Large double-blind placebo-controlled randomized clinical trial	Conversion to psychosis (the Comprehensive Assessment of theAt-Risk Mental State), functioning (Global Functioning: Social and Rolescales)	Conversion to psychosis status at 6 months for the intervention and control group was 6.7% and 5.1%, and at 12 months, 11.5% and 11.2%, respectively, suggesting no significant treatment effect of omega-3 PUFAs in reducing transition to psychosis. However, symptom and functional improvements were observed more broadly for both groups.
O’Kane, Sledjeski and Dinzeo	2022	Sleep	Northeastern USA	To investigate how sleep hygiene is related to three frequently examined sub-domains of schizotypy (i.e., positive, negative, disoganized) and QOL	Undergraduate students from a mid-sized university	N = 385 (F = 242, M = 143; *M_age_* = 20.83, *SD* = 3.61 years))	Self-report surveys; cross-sectional	Schizotypy (SPQ-BR), sleep hygiene (SHI)	Sleep hygiene (measured through patterns of pre-sleep behaviors) may be a relevant risk variable in the development of schizophrenia-spectrum symptoms. Sleep experiences associated with measures of schizotypy. Specifically, higher levels of schizotypy were associated with emotional rumination prior to sleep, while increased negative schizotypy was associated with poorer quality of life. Sex differences observed across schizotypy subscale except for disorganized features.
Orleans-Pobee, Browne, Ludwig et al.	2021	Exercise	Southeastern USA	To examine the impact of Physical Activity Can Enhance Life (PACE-Life), a novel walking intervention, on physical activity, and on secondary outcomes of cardiorespiratory fitness (CRF), physical health, autonomous motivation, social support, and quality of life	Individuals with SSDs were enrolled in a 10-week open trial	N = 16 (F = 2, M = 14)	Within-group effect sizes were calculated to represent changes from baseline to post-test and 1-month follow-up	Physical health, walking groups, home-based walks, Fitbit use, and goal setting and if-then plans (Fitbit PA trackers, IPAQ, BREQ-2, 6MWT)	Participants increased self-reported weekly walking minutes and decreased daily hours spent sitting; however, Fitbit-recorded exercise behavior changed only minimally. Cardiovascular fitness only improved marginally.
Pennacchi	2013	Nutrition and exercise	USA	Factors influencing health behaviors in those at risk for developing schizophrenia.	University students	N = 115 (F = 52%, M = 47%, other: 0.9%)	Self-report survey data, food diaries	Schizotypy (SPQ-B), nutrition (nutrition calculator), physical activity (GPAQ, LHQ-B), self-perception of fitness level relative to peers, eating habits (24-h diet recall log, interviews)	Only the unusual perceptual deficits subscale within the positive symptoms of schizotypy was predictive of healthier behaviors on health and exercise, and overall higher negative symptoms of schizotypy were associated with eating less nutritious foods (e.g., salad) and likelihood of eating out, self-perceived poorer fitness levels relative to peers, and with being less likely to engage in exercising. No relationship was found between negative schizotypy and eating habits.
Raine, Mellingen, Liu, Venables, and Mednick	2003	Nutrition	Mauritius, Africa	To examine the effects of environmental enrichment at ages 3–5 years on schizotypal personality and antisocial behavior at ages 17 and 23 years	3–5-year-olds in the treatment	N = 455 Children in treatment (N = 100) vs. control group (N = 355) (F = 48.6%, M = 51.4%)	Large double-blind placebo-controlled randomized trial	Schizotypy (SPQ), full behavioral outcome data were available at age 17 (n = 83) and 23 years (n = 73) that allowed for comparison with the control group (n = 355) Nutrition (7 indicators of malnutrition: reduced height for age, reduced weight for height, hemoglobin levels, angular stomatitis, hair dyspigmentation, thin hair, and hair that is easy to pull out)	This 2-year intervention found that compared to children in the control condition, no main effect for schizotypy and intervention groups at age 23—the effects were an interaction between enrichment and malnourished status at age 3. Main effects for schizotypy and intervention was found at age 17 years. These findings were especially stark for children who were deemed as malnourished at age 3 years, suggesting that an enriched, stimulating environment is beneficial for psychological and behavioral outcomes even 14 and 20 years later in life.
Reid and Zborowski	2006	Sleep	New York, USA	Investigated the birth timing effects during the year and how that is associated with levels of schizotypy and the amount of sleep.	Western New York University students	N = 452 (out of 530) (F = 75.4%, M = 24.6%; *M_age_* = 21.31, *SD* = 5.05 years)	Self-report surveys; cross-sectional	Birth date, schizotypy (PER-MAG), sleep	Individuals born in Winter rather than in Spring have higher schizotypy scores. Those sleeping <6 h or 8 h+ scored in the top 25% on the PER-MAG scale compared to those who slept 6,7,8 h. Higher schizotypy scores reported by those born in the Winter/Spring (21 Dec–20 June) > Summer/Fall (21 June–20 Dec).
Simor, Báthori, Nagy, and Polner	2019	Sleep	Hungarian	Poor sleep quality predicts psychotic-like symptoms: an experience sampling study in young adults with schizotypal traits	Hungarian university students with schizotypy	N = 73 (18–25 years old) (F = 61, M = 12)	Assessed subjective sleep quality and how it predicts fluctuations in PLEs the next few days and sleep quality.	Sleep quality (GSQS-H), psychotic-like experiences, schizotypy (s-OLIFE)	Found that poor sleep quality predicts psychotic-like experiences in young adults with schizotypy (n = 73 Hungarian university students 18–25 year-olds with moderate to high levels of positive schizotypy). Participants rated the sleep every morning, PLEs and affective stated during the day for 3 weeks. Participants with score > 4 on the Unusual Experience subscale (s-OLIFE). Found that subjective sleep quality predicts day-to-day fluctuations in PLEs in the next few days, poorer sleep quality and shorter sleep associated with increased PLEs the following day. Limitation no objective measure of sleep.
Watson	2001	Sleep	Iowa, USA	First, to create a simple, reliable measure tapping a broad range of sleep-related experiences. Second, to examine the nature of individual differences in sleep experiences. Third, to clarify the underlying structure of the broader domain by examining the nature of the relation between dissociation and schizotypy.	Two samples of students who were enrolled in an introductory psychology course at the University of Iowa	N = 482 Sample 1 (F = 285, M = 196, undefined = 1) N = 466 Sample 2 (F = 299, M = 166, undefined = 1)	Two large undergraduate samples self-reporting on a survey (Complete data on N_1_ = 471, N_2_ = 457)	Schizotypy (Perceptual Aberration scale, Magical Ideation scale, STQ Schizotypal Personality Scale) Sleep (ISES)	Moderate positive correlations between the Iowa Sleep Experience Survey (ISES) and self-report measures of dissociation and schizotypy—individuals who score high on the measures schizotypy also tend to score high on the ISES General Sleep Experiences Scale.
Wong, Wang, Esposito, & Raine	2022	Sleep	Worldwide	A three-wave network analysis of COVID-19’s impact on schizotypal traits, paranoia, and mental health through loneliness	General population adults aged 18–89 years-old primarily from the UK, US, Italy, and Greece across three time-points	Over 2300 participants (18–89 years) (F = 74.9%, M = 25.1%)	Self-report survey data	Sleep (Pittsburgh), schizotypy (SPQ-B)	Found that symptoms of schizotypy were more elevated at time 2 (October–January 2021) compared to times 1 and 3, and at every timepoint they were positively associated with poorer sleep, primarily through changes in the individual’s stress concerning the pandemic and depressive symptoms

## Data Availability

Data are included in this manuscript.

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
