# Peer review of "Nutrition, Sleep, and Exercise as Healthy Behaviors in Schizotypy: A Scoping Review"

_behavsci, 2022, doi:10.3390/bs12110412_

Round 1
Reviewer 1 Report
In the current manuscript author want to briefly discuss about the nutrition, sleep and exercise as a healthy behavior in Schizophrenia. However, the topic is very interesting, I have following major concern
1. This would be nice if author can provide recent data/reference about multidimensional disabling disorder affecting from Schizophrenia globally.
2. In the methodology section Records excluded (n= 19) however author has included 23 reviews, the number is mismatched, please check.
3. Venn Diagram is not good.
4. Et al., should be in italic.
5. Method and result section are poorly is poorly written.
6. Discussion section is also poorly described. Author did not briefly discuss about the persons having sleep and SPD or sleep, exercise and SPD and are under certain therapeutic drug interventions.
7. Author never discussed about the therapeutic interventions. If they intentionally not discussed about the therapeutic drug interventions they have not provided a proper justification .
Reviewer 2 Report
The article does not present the prism record for revisions;
There are no training control parameters that justify the prescription and control of exercise, thus leading to a scientific mistake in the production of the manuscript.
Reviewer 3 Report
The work does not meet the minimum methodological quality for publication.
The limited number of articles in each variable are not significant for drawing conclusions.
Reviewer 4 Report
In attachment

Reviewer 5 Report
This scoping review focused on the use of three adaptive behaviors in schizotypy: nutrition, sleep, and physical activity. They introduced that there was some evidence that those with schizotypy exhibit poorer health behaviors in these three areas but that a scoping review could outline these findings more clearly and provide recommendations for future research. The review made few conclusive findings due to a dearth of studies in the field and suggested the promotion of intersectional studies to examine these behaviors more clearly in the future. Strengths of this review include its importance, the clear writing, and the growing interest in adaptive behaviors across the psychosis-spectrum. However, there were also some concerns that limited my enthusiasm for the manuscript. Please consider these comments as you revise this manuscript for this or another journal.
1. The largest issue with the manuscript is one that the authors will not be able to address: there is a lack of conclusive findings across constructs of interest. In many ways, the review does not expand past the initial findings outlined in the introduction. Could the authors summarize each of these areas and describe how their review expands on the knowledge they had coming into the review?
2. As the authors note in the conclusion, it is important to examine specific dimensions of schizotypy in addition to schizotypy as its own group. However, little has been done in terms of testing how positive, negative, and disorganized traits are associated with the constructs of interest here. Are there recommendations for how future studies should structure their projects to obtain these answers?
3. A stronger focus on recommendations would also help improve the review. Given the lack of answers, having key takeaway seems particularly important for this review. In line with this, what are the barriers that people with schizotypy face in implementing author recommendations? For example, the authors note that ‘prescribing’ physical activity for those with SSD might be a solution. However, all of us face barriers in increasing our physical activity and these barriers are salient for those with SSD. What are these barriers and how can those with SSD circumvent them?
4. Were there any cultural effects noticed in the findings? Studies originated all over the world and I wonder if there are any cultural considerations that might impact people differently across studies?
5. Method: Did the authors use ‘Schizotypal Personality Disorder’ in place of ‘schizotypy’ and ‘schizotypal’ or in addition to these search terms? The latter approach is how I read this but the former approach could lead to problems identifying all relevant studies.
6. Small point: The authors note 42 articles originally were considered but Figure 1 lists 41. There were also a few other typos or incorrect word choices in the manuscript.
Reviewer 6 Report
Thank you for the opportunity to review this scoping review paper nutrition, sleep, and healthy behaviors in schizotypy. I commend the authors for their attention toward health outcomes in this population – these are very important and largely understudies aspects of psychological function and well-being among people at risk for psychosis. Overall the rationale for a scoping review (vs a meta-analysis or systematic review) is appropriate and justified by the authors. At the same time, I have some concerns about how schizotypy, schizotypal personality disorder, and psychotic disorders are presented somewhat interchangeably throughout the paper and related impact on inferences that can be drawn in the current version. I will list specific comments below, and I hope they will assist the authors in considering next steps for this work.
Overall Comments:
11) In the introduction, the authors first introduce schizophrenia and schizophrenia-spectrum disorders, and then lay out how schizotypal personality disorder (SPD) is related to this larger spectrum of psychosis/psychosis-proneness. The term schizotypy is then used but never defined, which is a crucial area for clarification given the papers that are included in the review. I suggest that the authors revise the introduction to introduce the concept of psychometric schizotypy and how this is different/similar to the clinical diagnosis of SPD.
22) Without further clarification on how these disorders/clinical states intersect, in its current version the paper does not read as a review about SPDs but more a review on SMI more broadly.
Nutrition & SPD section:
13) On pg. 5 the authors note a finding of a “significant main effect for schizotypy and intervention.” This should be explained more clearly – is the main effect for a particular aspect/dimension of schizotypy? With risk for developing SPD? Readers without statistical/ANOVA familiarity are not likely to understand what main effect means here.
24) On like 235, the authors note “experimentally robust evidence” for nutritional enrichment on reducing psychotic symptoms and other outcomes – given that some of the other papers discussed in this section had null findings (McGorry et al) or were not experimental (2006 interviews/diary recreation), this may not be the most appropriate phrasing. I agree that additional research is warranted, and perhaps that can be a more central focus of this paragraph.
Exercise & SPD:
15) This comment relates to my overall comment about clarifying SDP vs schizotypal vs SSDs. From my read of this section, I don’t believe the authors review any data from individuals with SPDs? Two of the papers focus on established psychotic disorders, with one UHR paper that has unclear SPD representation.
Sleep & SPD
16) This section does not present data on SPDs but instead largely data from college populations with psychometrically-defined schizotypy.
Discussion
17) I appreciate the author statement about the need for qualitative data to improve our understanding of health behaviors in this population – very important but often overlooked.
28) Specific mention of limitations should appear here. As the authors revise the manuscript to better describe SPDs vs schizotypy vs psychotic disorders, information on how combining data across all of these groups impacts/limits inferences that can be made will likely be appropriate.
Round 2
Reviewer 1 Report
Manuscript has been revised as per the concern.
Author Response
Thanks for taking the time to review our manuscript.
Reviewer 3 Report
The work has NOT been substantially improved for publication (they use PRISMA and do not cite it; It is not relevant to eliminate articles due to not having access to the full text, etc.).
Author Response
Thanks for spotting this. I have now included the reference for PRISMA (reference 16) and uploaded the PRISMA checklist onto OSF so readers can access this. As for eliminating articles that cannot be accessible, this was based on the example set in the PRISMA statement (https://www.acpjournals.org/doi/10.7326/M18-0850) where the authors excluded the study because it cannot be retrieved. We did not feel confident to include an article we could not access completely.
Thanks for taking the time to review our manuscript.
Reviewer 4 Report
I thank the authors for the changes made. The manuscript has improved substantially.
Author Response
We thank you for taking the time to review our manuscript.
Reviewer 5 Report
The authors adequately responded to my comments. There are still some issues with the review but many of them are not areas that can be corrected (i.e., lack of conclusive findings).
Author Response

(The authors gave the same response as above.)

Reviewer 6 Report
author responses to my previous comments are acceptable
Author Response

(The authors gave the same response as above.)
